# Isotope analysis in the transmission electron microscope

Toma Susi[1], Christoph Hofer[1], Giacomo Argentero[1], Gregor T. Leuthner[1], Timothy J. Pennycook[1], Clemens Mangler[1], Jannik C. Meyer[1] & Jani Kotakoski[1]

The Ångström-sized probe of the scanning transmission electron microscope can visualize and collect spectra from single atoms. This can unambiguously resolve the chemical structure of materials, but not their isotopic composition. Here we differentiate between two isotopes of the same element by quantifying how likely the energetic imaging electrons are to eject atoms. First, we measure the displacement probability in graphene grown from either $^{12}$C or $^{13}$C and describe the process using a quantum mechanical model of lattice vibrations coupled with density functional theory simulations. We then test our spatial resolution in a mixed sample by ejecting individual atoms from nanoscale areas spanning an interface region that is far from atomically sharp, mapping the isotope concentration with a precision better than 20%. Although we use a scanning instrument, our method may be applicable to any atomic resolution transmission electron microscope and to other low-dimensional materials.

[1] Faculty of Physics, University of Vienna, Faculty of Physics, Boltzmanngasse 5, 1090 Vienna, Austria. Correspondence and requests for materials should be addressed to T.S. (email: toma.susi@univie.ac.at) or to J.K. (email: jani.kotakoski@univie.ac.at).

Spectroscopy and microscopy are two fundamental pillars of materials science. By overcoming the diffraction limit of light, electron microscopy has emerged as a particularly powerful tool for studying low-dimensional materials such as graphene[1], in which each atom can be distinguished. Through advances in aberration-corrected scanning transmission electron microscopy[2,3] (STEM) and electron energy loss spectroscopy[4,5], the vision of a 'synchrotron in a microscope'[6] has now been realized. Spectroscopy of single atoms, including their spin state[7], has together with Z-contrast imaging[3] allowed the identity and bonding of individual atoms to be unambiguously determined[4,8–10]. However, discerning the isotopes of a particular element has not been possible—a technique that might be called 'mass spectrometer in a microscope'.

Here we show how the quantum mechanical description of lattice vibrations lets us accurately model the stochastic ejection of single atoms[11,12] from graphene consisting of either of the two stable carbon isotopes. Our technique rests on a crucial difference between electrons and photons when used as a microscopy probe: due to their finite mass, electrons can transfer significant amounts of momentum. When a highly energetic electron is scattered by the electrostatic potential of an atomic nucleus, a maximal amount of kinetic energy (inversely proportional to the mass of the nucleus, $\propto \frac{1}{M}$) can be transferred when the electron backscatters. When this energy is comparable to the energy required to eject an atom from the material, defined as the displacement threshold energy $T_d$—for instance, when probing pristine[11] or doped[13] single-layer graphene with 60–100 keV electrons—atomic vibrations become important in activating otherwise energetically prohibited processes due to the motion of the nucleus in the direction of the electron beam. The intrinsic capability of STEM for imaging further allows us to map the isotope concentration in selected nanoscale areas of a mixed sample, demonstrating the spatial resolution of our technique. The ability to do mass analysis in the transmission electron microscope thus expands the possibilities for studying materials on the atomic scale.

## Results

**Quantum description of vibrations.** The velocities of atoms in a solid are distributed based on a temperature-dependent velocity distribution, defined by the vibrational modes of the material. Due to the geometry of a typical transmission electron microscopy (TEM) study of a two-dimensional material, the out-of-plane velocity $v_z$, whose distribution is characterized by the mean square velocity $\overline{v_z^2}(T)$, is here of particular interest. In an earlier study[11] this was estimated using a Debye approximation for the out-of-plane phonon density of states[14] (DOS) $g_z(\omega)$, where $\omega$ is the phonon frequency. A better justified estimate can be achieved by calculating the kinetic energy of the atoms via the thermodynamic internal energy, evaluated using the full phonon DOS.

As a starting point, we calculate the partition function $Z = \mathrm{Tr}\{e^{-H/(kT)}\}$, where Tr denotes the trace operation and $k$ is the Boltzmann constant and $T$ the absolute temperature. We evaluate this trace for the second-quantized Hamiltonian $H$ describing harmonic lattice vibrations[15]:

$$Z = \sum_{n_{j_1}(\mathbf{k}_1)=0}^{\infty} ... \sum_{n_{j_{3r}}(\mathbf{k}_N)=0}^{\infty} \exp\left(-\frac{1}{kT}\sum_{\mathbf{k}j}\hbar\omega_j(\mathbf{k})\left[n_j(\mathbf{k})+\frac{1}{2}\right]\right)$$
$$= \prod_{\mathbf{k}j}\frac{\exp\left(-\frac{1}{2}\hbar\omega_j(\mathbf{k})/(kT)\right)}{1-\exp\left(-\hbar\omega_j(\mathbf{k})/(kT)\right)}, \quad (1)$$

where $\hbar$ is the reduced Planck constant, $\mathbf{k}$ the phonon wave vector, $j$ the phonon branch index running to $3r$ ($r$ being the

number of atoms in the unit cell), $\omega_j(\mathbf{k})$ the eigenvalue of the $j_{th}$ mode at $\mathbf{k}$, and $n_j(\mathbf{k})$ the number of phonons with frequency $\omega_j(\mathbf{k})$.

After computing the internal energy $U = F - T\left(\frac{\partial F}{\partial T}\right)_V$ from the partition function via the Helmholtz free energy $F = -kT \ln Z$, we obtain the Planck distribution function describing the occupation of the phonon bands (Methods). We must then explicitly separate the energy into the in-plane $U_p$ and out-of-plane $U_z$ components, and take into account that half the thermal energy equals the kinetic energy of the atoms. This gives the out-of-plane mean square velocity of a single atom in a two-atom unit cell as

$$\overline{v_z^2}(T) = U_z/(2M) = \frac{\hbar}{2M}\int_0^{\omega_z} g_z(\omega)\left[\frac{1}{2}+\frac{1}{\exp(\hbar\omega/(kT))-1}\right]\omega\,d\omega, \quad (2)$$

where $M$ is the mass of the vibrating atom, $\omega_z$ is the highest out-of-plane mode frequency, and the correct normalization of the number of modes $\left(\int_0^{\omega_z} g_z(\omega)\,d\omega = 2\right)$ is included in the DOS.

**Phonon dispersion.** To estimate the phonon DOS, we calculated through density functional theory (DFT; *GPAW* package[16,17]) the graphene phonon band structure[18,19] via the dynamical matrix using the 'frozen phonon method' (Methods; Supplementary Fig. 1). Taking the density of the components corresponding to the out-of-plane acoustic (ZA) and optical (ZO) phonon modes (Supplementary Data 1) and solving equation 2 numerically, we obtain a mean square velocity $\overline{v_z^2} \approx 3.17 \times 10^5$ m$^2$s$^{-2}$ for a $^{12}$C atom in normal graphene. This description can be extended to 'heavy graphene' (consisting of $^{13}$C instead of a natural isotope mixture). A heavier atomic mass affects the velocity through two effects: the phonon band structure is scaled by the square root of the mass ratio (from the mass prefactor of the dynamical matrix), and the squared velocity is scaled by the mass ratio itself (equation 2). At room temperature, the first correction reduces the velocity by 3% in fully $^{13}$C graphene compared with normal graphene, and the second one reduces it by an additional 10%, resulting in $\overline{v_{z,13}^2} \approx 2.86 \times 10^5$ m$^2$s$^{-2}$.

**Electron microscopy.** In our experiments, we recorded time series at room temperature using the Nion UltraSTEM100 microscope, where each atom, or its loss, was visible in every frame. We chose small fields of view ($\sim 1 \times 1$ nm$^2$) and short dwell times (8 µs) to avoid missing the refilling of vacancies (an example is shown in Fig. 1; likely this vacancy only appears to be unreconstructed due to the scanning probe). In addition to commercial monolayer graphene samples (Quantifoil R 2/4, Graphenea), we used samples of $^{13}$C graphene synthesized by chemical vapour deposition (CVD) on Cu foils using $^{13}$C-substituted CH$_4$ as carbon precursor, subsequently transferred onto Quantifoil TEM grids. An additional sample consisted of grains of $^{12}$C and $^{13}$C graphene on the same grid, synthesized by switching the precursor during growth (Methods).

From each experimental dataset (full STEM data available[20]) within which a clear displacement was observed, we calculated the accumulated electron dose until the frame where the defect appeared (or a fraction of the frame if it appeared in the first one). The distribution of doses corresponds to a Poisson process[12] whose expected value was found by log-likelihood minimization (Methods; Supplementary Fig. 2), directly yielding the probability of creating a vacancy (the dose data and statistical analyses are included in Supplementary Data 2). Figure 2 displays the corresponding displacement cross sections measured at voltages between 80 and 100 kV for normal (1.109% $^{13}$C) and heavy graphene ($\sim 99\%$ $^{13}$C), alongside values measured earlier[11] using

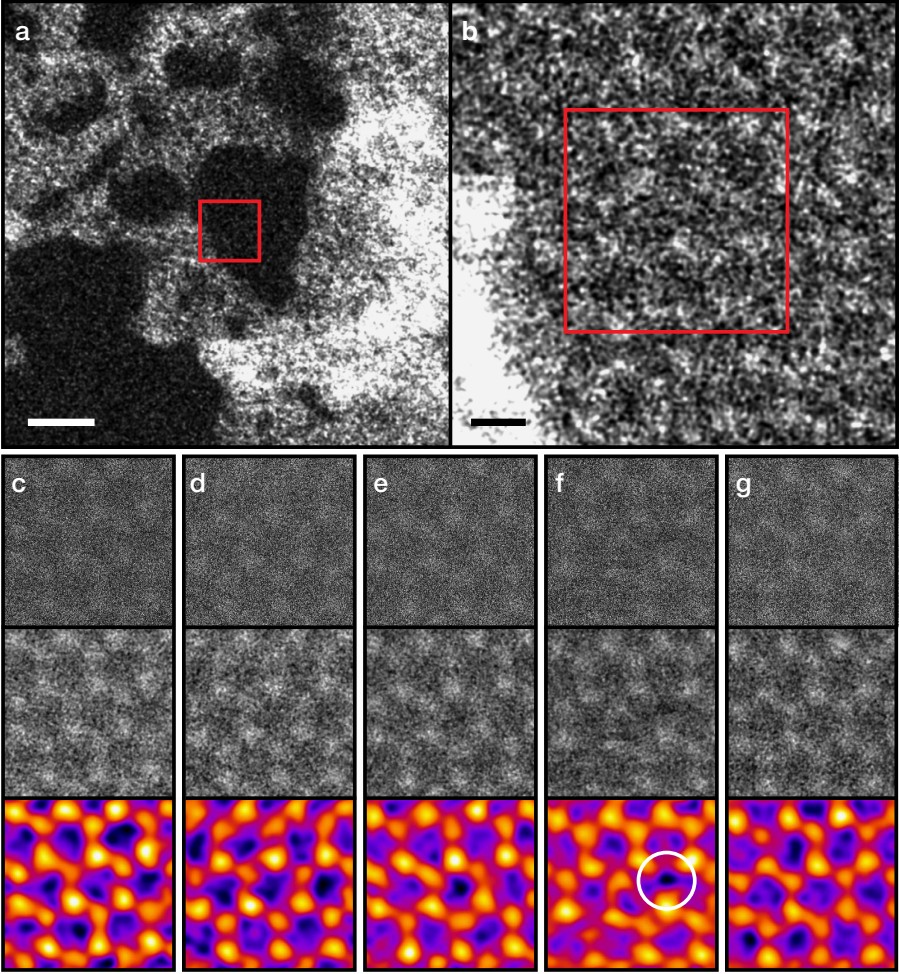

**Figure 1 | Example of the STEM displacement measurements.** The micrographs are medium angle annular dark field detector images recorded at 95 kV. (**a**) A spot on the graphene membrane, containing clean monolayer graphene areas (dark) and overlying contamination (bright). Scale bar, 2 nm. (**b**) A closer view of the area marked by the red rectangle in (**a**), with the irradiated area of the following panels similarly denoted. Scale bar, 2 Å. (**c–g**) Five consecutive STEM frames ($\sim 1 \times 1\,nm^2$, $512 \times 512$ pixels (px), 2.2 s per frame) recorded at a clean monolayer area of graphene. A single carbon atom has been ejected in the fourth frame (**f**, white circle), but the vacancy is filled already in the next frame (**g**). The top row of (**c–g**) contains the unprocessed images, the middle row has been treated by a Gaussian blur with a radius of 2 px, and the coloured bottom row has been filtered with a double Gaussian procedure[3] ($\sigma_1 = 5$ px, $\sigma_2 = 2$ px, weight $= 0.16$).

high-resolution TEM (HRTEM). For low-probability processes, the cross section is highly sensitive to both the atomic velocities and the displacement threshold energy. Since heavier atoms do not vibrate with as great a velocity, they receive less of a boost to the momentum transfer from an impinging electron. Thus, fewer ejections are observed for $^{13}C$ graphene.

**Comparing theory with experiment**. The theoretical total cross sections $\sigma_d(T, E_e)$ are plotted in Fig. 2 for each voltage (Methods; Supplementary Table 1, Supplementary Data 2). The motion of the nuclei was included via a Gaussian distribution of atomic out-of-plane velocities $P(v_z, T)$ characterized by the DFT-calculated $\overline{v_z^2}$, otherwise similar to the approach of ref. 11. A common displacement threshold energy was fitted to the data set by minimizing the variance-weighted mean square error (the 100 kV HRTEM point was omitted from the fitting, since it was underestimated probably due to the undetected refilling of vacancies, also seen in Fig. 1). The optimal $T_d$ value was found to be 21.14 eV, resulting in a good description of all the measured cross sections. Notably, this is 0.8 eV lower than the earlier value calculated by DFT, and 2.29 eV lower than the earlier fit to HRTEM data[11]. Different exchange correlation functionals we

tested all overestimate the experimental value (by $< 1$ eV), with the estimate $T_d \in [21.25, 21.375]$ closest to experiment resulting from the C09 van der Waals functional[21] (Methods).

Despite DFT overestimating the displacement threshold energy, we see from the good fit to the normal and heavy graphene data sets that our theory accurately describes the contribution of vibrations. Further, the HRTEM data and the STEM data are equally well described by the theory despite having several orders of magnitude different irradiation dose rates. This can be understood in terms of the very short lifetimes of electronic and phononic excitations in a metallic system[22] compared with the average time between impacts. Even a very high dose rate of $10^8\,e^- \AA^{-2}s^{-1}$ corresponds to a single electron passing through a 1 nm$^2$ area every $10^{-10}$ s, whereas valence band holes are filled[23] in $< 10^{-15}$ s and core holes[24] in $< 10^{-14}$ s, while plasmons are damped[25] within $\sim 10^{-13}$ s and phonons[26] in $\sim 10^{-12}$ s. Our results thus show that multiple excitations do not contribute to the knock-on damage in graphene, warranting another explanation (such as chemical etching[11]) for the evidence linking a highly focused HRTEM beam to defect creation[27]. Each impact is, effectively, an individual perturbation of the equilibrium state.

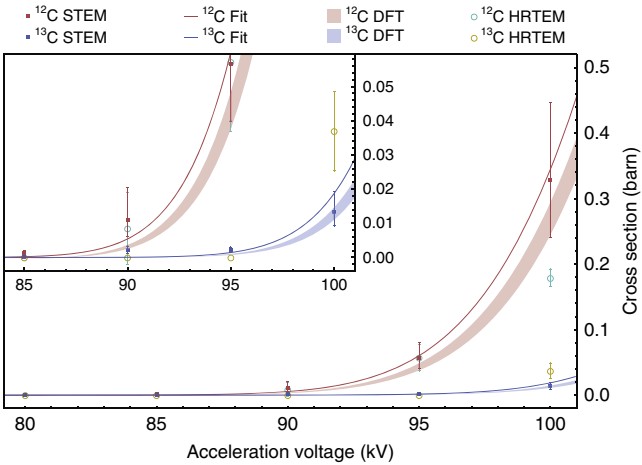

**Figure 2 | Displacement cross sections of $^{12}$C and $^{13}$C measured at different acceleration voltages.** The STEM data is marked with squares, and earlier HRTEM data[11] with circles. The error bars correspond to the 95% confidence intervals of the Poisson means (STEM data) or to previously reported estimates of statistical variation (HRTEM data[11]). The solid curves are derived from our theoretical model with an error-weighted least-squares best-fit displacement threshold energy of 21.14 eV. The shaded areas correspond to the same model using the lowest DFT threshold $T_d \in [21.25, 21.375]$ eV. The inset is a closer view of the low cross section region.

**Local mapping of isotope concentration.** Finally, to test the spatial resolution of our method, we studied a sample consisting of joined grains of $^{12}$C and $^{13}$C graphene. Isotope labelling combined with Raman spectroscopy mapping is a powerful tool for studying CVD growth of graphene[28], which is of considerable technological interest. Earlier studies have revealed the importance of carbon solubility into different catalyst substrates to control the growth process[29]. However, the spatial resolution of Raman spectroscopy is limited, making it impossible to obtain atomic-scale information of the transition region between grains of different isotopes.

The local isotope analysis is based on fitting the mean of the locally measured electron doses with a linear combination of doses generated by Poisson processes corresponding to $^{12}$C and $^{13}$C graphene using the theoretical cross section values. Although each dose results from a stochastic process, the expected doses for $^{12}$C and $^{13}$C are sufficiently different that measuring several displacements decreases the errors of their means well below the expected separation (Fig. 3c). To estimate the expected statistical variation for a certain number of measured doses, we generated a large number of sets of $n$ Poisson doses, and calculated their means and standard errors as a function of the number of doses in each set. The calculated relative errors scale as $1/n$ and correspond to the precision of our measurement, which is better than 20% for as few as five measured doses in the ideal case. Although our accuracy is difficult to gauge precisely, by comparing the errors of the cross sections measured for isotopically pure samples to the fitted curve (Fig. 2), an estimate of roughly 5% can be inferred.

Working at 100 kV, we selected spots containing areas of clean graphene (43 in total) each only a few tens of nanometers in size (Fig. 1a), irradiating 4–15 (mean 7.8) fields of view $1 \times 1$ nm$^2$ in size until the first displacement occurred (Fig. 1f). Comparing the mean of the measured doses to the generated data, we can estimate the isotope concentration responsible for such a dose. This assignment was corroborated by Raman mapping over the same area, allowing the two isotopes to be distinguished by their

differing Raman shift. A general trend from $^{12}$C-rich to $^{13}$C-rich regions is captured by both methods (Fig. 3b), but a significant local variation in the measured doses is detectable (Fig. 3c). This variation indicates that the interfaces formed in a sequential CVD growth process may be far from atomically sharp[30], instead spanning a region of hundreds of nanometers, within which the carbon isotopes from the two precursors are mixed together.

## Discussion

It is interesting to compare our method to established mass analysis techniques. In isotope ratio mass spectrometry precisions of 0.01% and accuracies of 1% have been reported[31]. However, these measurements are not spatially resolved. For spatially resolved techniques, one of the most widely used is time-of-flight secondary ion mass spectroscopy (ToF-SIMS). It has a lateral resolution typically of several micrometers, which can be reduced to around 100 nm by finely focusing the ion beam[32]. In the case of ToF-SIMS, separation of the $^{13}$C signal from $^{12}$C$^1$H is problematic, resulting in a reported[33] precision of 20% and an accuracy of $\sim$11%. The state-of-the-art performance in local mass analysis can be achieved with atom-probe tomography[34] (APT), which can record images with sub-nanometer spatial resolution in all three dimensions. A recent APT study of the $^{13}$C/$^{12}$C ratio in detonation nanodiamonds reported a precision of 5%, but biases in the detection of differently charged ions limited accuracy to $\sim$25% compared to the natural isotope abundances[35].

A limitation of ToF-SIMS is its inability to discriminate between the analyte and contaminants and that it requires uniform isotope concentrations over the beam area for accurate results. APT requires the preparation of specialized needle-like sample geometries, a laborious reconstruction process to analyse its results[36], and its detection efficiency is rather limited[37]. In our case, we are only able to resolve relative mass differences between isotopes of the same element in the same chemical environment. While we do not need to resolve mass differences between different elements, since these differ in their scattering contrast, we do need to detect the ejection of single atoms, limiting the technique to atomically thin materials. However, our method captures the isotope information concurrently with atomic resolution imaging in a general-purpose electron microscope, without the need for additional detectors.

We have shown how the Ångström-sized electron probe of a scanning transmission electron microscope can be used to estimate isotope concentrations via the displacement of single atoms. Although these results were achieved with graphene, our technique should work for any low-dimensional material, including hexagonal boron nitride and transition metal dichalcogenides such as MoS$_2$. This could potentially extend to van der Waals heterostructures[38] of a few layers or other thin crystalline materials, provided a difference in the displacement probability of an atomic species can be uniquely determined. Neither is the technique limited to STEM: a parallel illumination TEM with atomic resolution would also work, although scanning has the advantage of not averaging the image contrast over the field of view. The areas we sampled were in total less than 340 nm$^2$ in size, containing $\sim$6,600 carbon atoms of which 337 were ejected. Thus, while the nominal mass required for our complete analysis was already extremely small (131 zg), the displacement of only five atoms is required to distinguish a concentration difference of less than twenty per cent. Future developments in instrumentation may allow the mass-dependent energy transfer to be directly measured from high-angle scattering[39,40], further enhancing the capabilities of STEM for isotope analysis.

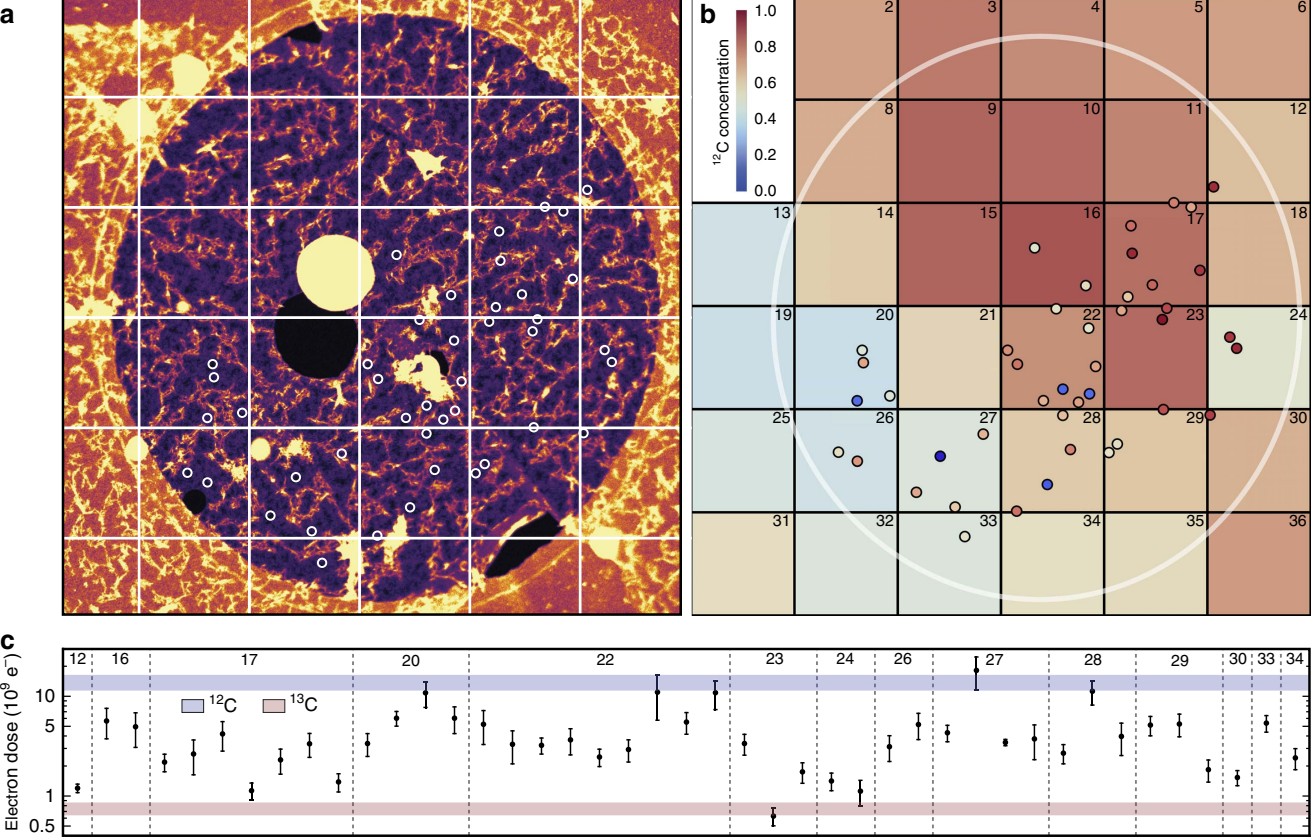

**Figure 3 | Local isotope analysis. (a)** A STEM micrograph of a hole in the carbon support film (1.3 µm in diameter), covered by a monolayer of graphene. In each of the overlaid spots, 4–15 fields of view were irradiated. The dimensions of the overlaid grid correspond to the pixels of a Raman map recorded over this area. **(b)** Isotope concentration map where the colours of the grid squares denote $^{12}$C concentration based on the fitting of the Raman 2D band response (Methods; Supplementary Fig. 3). The overlaid spots correspond to **(a)**, with colours denoting the concentration of $^{12}$C estimated from the mean of the measured doses. **(c)** Locally measured mean doses and their standard errors plotted on a log scale for each grid square. The horizontal coloured areas show the means ± s.e. of doses simulated for the theoretical $^{12}$C and $^{13}$C cross sections. Note that a greater variation in the experimental doses is expected for areas containing a mix of both carbon isotopes.

## Methods

**Quantum model of vibrations.** The out-of-plane mean square velocity $\overline{v_z^2}(T)$ can be estimated by calculating the kinetic energy via the thermodynamic internal energy using the out-of-plane phonon DOS $g_z(\omega)$, where $\omega$ is the phonon frequency. In the second quantization formalism, the Hamiltonian for harmonic lattice vibrations is ref. 15

$$H = \sum_{\mathbf{k}}^{N} \sum_{j=1}^{3r} \hbar\omega_j(\mathbf{k}) \left[ b_{\mathbf{k}j}^{\dagger} b_{\mathbf{k}j} + \frac{1}{2} \right], \quad (3)$$

where $\mathbf{k}$ is the phonon wave vector, $j$ is the phonon branch index running to $3r$ ($r$ being the number of atoms in the unit cell), $\omega_j(\mathbf{k})$ the eigenvalue of the $j$th mode at $\mathbf{k}$, and $b_{\mathbf{k}j}^{\dagger}$ and $b_{\mathbf{k}j}$ are the phonon creation and annihilation operators, respectively.

Using the partition function $Z = \text{Tr}\{e^{-H/(kT)}\}$, where Tr denotes the trace operation and $k$ is the Boltzmann constant and $T$ the absolute temperature, and evaluating the trace using this Hamiltonian, we have

$$Z = \sum_{n_{j_1}(\mathbf{k}_1)=0}^{\infty} \dots \sum_{n_{j_{3r}}(\mathbf{k}_N)=0}^{\infty} \exp\left( -\frac{1}{kT} \sum_{\mathbf{k}j} \hbar\omega_j(\mathbf{k}) \left[ n_j(\mathbf{k}) + \frac{1}{2} \right] \right)$$
$$= \prod_{\mathbf{k}j} \frac{\exp\left( -\frac{1}{2}\hbar\omega_j(\mathbf{k})/(kT) \right)}{1 - \exp\left( -\hbar\omega_j(\mathbf{k})/(kT) \right)}, \quad (4)$$

where $n_j(\mathbf{k}) = b_{\mathbf{k}j}^{\dagger} b_{\mathbf{k}j}$ is the number of phonons with frequency $\omega_j(\mathbf{k})$.

The Helmholtz free energy is thus

$$F = -kT \ln Z = kT \sum_{\mathbf{k}j} \ln\left[ 2\sinh\left( \hbar\omega_j(\mathbf{k})/(2kT) \right) \right] \quad (5)$$

and the internal energy of a single unit cell, therefore, becomes[15]

$$U = F - T\left( \frac{\partial F}{\partial T} \right)_V = \sum_{\mathbf{k}j} \frac{1}{2}\hbar\omega_j(\mathbf{k})\coth\left( \hbar\omega_j(\mathbf{k})/(2kT) \right)$$
$$= 3r \int \frac{1}{2}\coth(\hbar\omega/(2kT))g(\omega)\hbar\omega\,d\omega, \quad (6)$$

where in the last step the sum is expressed as an average over the phonon DOS. Using the identity $\frac{1}{2}\coth(x/2) = \frac{1}{2} + 1/(\exp(x)-1)$ yields the Planck distribution function describing the occupation of the phonon bands, and explicitly dividing the energy into the in-plane $U_p$ and out-of-plane $U_z$ components, we can rewrite this as

$$U = U_p + U_z = \int_0^{\omega_d} \left( g_p(\omega) + g_z(\omega) \right) \left[ \frac{1}{2} + \frac{1}{\exp(\hbar\omega/(kT))-1} \right] \hbar\omega\,d\omega, \quad (7)$$

where the number of modes is included in the normalization of the DOSes, that is, $\int_0^{\omega_z} g_z(\omega)d\omega = 2$, corresponding to the out-of-plane acoustic (ZA) and optical (ZO) modes (the in-plane DOS $g_p(\omega)$ being correspondingly normalized to 4), and $\omega_d$ is the highest frequency of the highest phonon mode.

Since half of the thermal energy equals the average kinetic energy of the atoms, and the graphene unit cell has two atoms, the out-of-plane kinetic energy of a single atom is

$$E_{k,z} = \frac{1}{2}M\overline{v_z^2} = \frac{1}{2}\frac{1}{2}U_z. \quad (8)$$

Thus, the out-of-plane mean square velocity of an atom becomes

$$\overline{v_z^2}(T) = U_z/(2M) = \frac{\hbar}{2M} \int_0^{\omega_z} g_z(\omega)\left[ \frac{1}{2} + \frac{1}{\exp(\hbar\omega/(kT))-1} \right] \omega\,d\omega, \quad (9)$$

where $\omega_z$ is now the highest out-of-plane mode frequency. This can be solved numerically for a known $g_z(\omega)$.

For the in-plane vibrations, we would equivalently get

$$\overline{v_{\mathrm{p}}^2} = \overline{v_{\mathrm{x}}^2} + \overline{v_{\mathrm{y}}^2} = U_{\mathrm{p}}/(2M) = \frac{\hbar}{2M} \int_0^{\omega_{\mathrm{p}}} g_{\mathrm{p}}(\omega) \left[ \frac{1}{2} + \frac{1}{\exp(\hbar\omega/(kT)) - 1} \right] \omega \, d\omega. \quad (10)$$

**Frozen phonon calculation.** To estimate the phonon DOS, we calculated the graphene phonon band structure via the dynamical matrix, which was computed by displacing each of the two primitive cell atoms by a small displacement (0.06 Å) and calculating the forces on all other atoms in a $7 \times 7$ supercell ('frozen phonon method'; the cell size is large enough so that the forces on the atoms at its edges are negligible) using DFT as implemented in the grid-based projector-augmented wave code (GPAW) package[17]. Exchange and correlation were described by the local density approximation[41], and a $\Gamma$-centered Monkhorst-Pack **k**-point mesh of $42 \times 42 \times 1$ was used to sample the Brillouin zone. A fine computational grid spacing of 0.14 Å was used alongside strict convergence criteria for the structural relaxation (forces $< 10^{-5}$ eVÅ$^{-1}$ per atom) and the self-consistency cycle (change in eigenstates $< 10^{-13}$ eV$^2$ per electron). The resulting phonon dispersion (Supplementary Fig. 1) describes well the quadratic dispersion of the ZA mode near $\Gamma$, and is in excellent agreement with earlier studies[18,19]. Supplementary Data 1 contains the out-of-plane phonon DOS.

**Graphene synthesis and transfer.** In addition to commercial monolayer graphene (Graphenea QUANTIFOIL R 2/4), our graphene samples were synthesized by CVD in a furnace equipped with two separate gas inlets that allow for independent control over the two isotope precursors[29] (that is, either $\sim 99\%$ $^{12}$CH$_4$ or $\sim 99\%$ $^{13}$CH$_4$ methane). The as-received 25 µm thick 99.999% pure Cu foil was annealed for $\sim 1$ h at $960\,^\circ$C in a 1:20 hydrogen/argon mixture with a pressure of $\sim 10$ mbar. The growth of graphene was achieved by flowing 50 cm$^3$ min$^{-1}$ of CH$_4$ over the annealed substrate while keeping the Ar/H$_2$ flow, temperature and pressure constant. For the isotopically mixed sample with separated domains, the annealing and growth temperature was increased to 1,045 °C and the flow rate decreased to 2 cm$^3$ min$^{-1}$. After introducing $^{12}$CH$_4$ for 2 min the carbon precursor flow was stopped for 10 s, and the other isotope precursor subsequently introduced into the chamber for another 2 min. This procedure was repeated with a flow time of 1 min. After the growth, the CH$_4$ flow was interrupted and the heating turned off, while the Ar/H$_2$ flow was kept unchanged until the substrate reached room temperature. The graphene was subsequently transferred onto a holey amorphous carbon film supported by a TEM grid using a direct transfer method without using polymer[42].

**Scanning transmission electron microscopy.** Electron microscopy experiments were conducted using a Nion UltraSTEM100 scanning transmission electron microscope, operated between 80 and 100 kV in near-ultrahigh vacuum ($2 \times 10^{-7}$ Pa). The instrument was aligned for each voltage so that atomic resolution was achieved in all of the experiments. The beam current during the experiments varied between 8 and 80 pA depending on the voltage, corresponding to dose rates of $\sim 5$–$50 \times 10^7$ e$^-$ Å$^{-2}$s$^{-1}$. The beam convergence semiangle was 30 mrad and the semi-angular range of the medium-angle annular-dark-field detector was 60–200 mrad.

**Poisson analysis.** Assuming the displacement data are stochastic, the waiting times (or, equivalently, the doses) should arise from a Poisson process with mean $\lambda$. Thus the probability to find $k$ events in a given time interval follows the Poisson distribution

$$f(k; \lambda) = \Pr(X = k) = \frac{\lambda^k e^{-\lambda}}{k!}. \quad (11)$$

To estimate the Poisson expectation value for each sample and voltage, the cumulative doses of each data set were divided into bins of width $w$ (using one-level recursive approximate Wand binning[43]), and the number of bins with 0, 1, 2... occurrences were counted. The goodness of the fits was estimated by calculating the Cash C-statistic[44] (in the asymptotically-$\chi^2$ formulation[45]) between a fitted Poisson distribution and the data:

$$C = \frac{2}{N} \sum_{i=1}^{N} \left[ n_i \ln \frac{n_i}{e_i} - (n_i - e_i) \right], \quad (12)$$

where $N$ is the number of occurence bins, $n_i$ is the number of events in bin $i$, and $e_i$ is the expected number of events in bin $i$ from a Poisson process with mean $\lambda$.

An error estimate for the mean was calculated using the approximate confidence interval proposed for Poisson processes with small means and small sample sizes by Khamkong[46]:

$$CI_{95\%} = \bar{\lambda} + \frac{Z_{2.5}^2}{2n} \pm Z_{2.5} \sqrt{\frac{\bar{\lambda}}{n}}, \quad (13)$$

where $\bar{\lambda}$ is the estimated mean and $Z_{2.5}$ is the normal distribution single tail cumulative probability corresponding to a confidence level of $(100 - \alpha) = 95\%$, equal to 1.96.

The statistical analyses were conducted using the Wolfram Mathematica software (version 10.5), and the Mathematica notebook is included as Supplementary Data 2. Outputs of the Poisson analyses for the main data sets of normal and heavy graphene as a function of voltage are additionally shown as Supplementary Fig. 2.

**Displacement cross section.** The energy transferred to an atomic nucleus from a fast electron as a function of the electron scattering angle $\theta$ is ref. 47

$$E(\theta) \approx E_{\max} \sin^2\left(\frac{\theta}{2}\right), \quad (14)$$

which is valid also for a moving target nucleus for electron energies $> 10$ keV as noted by Meyer and co-workers[11]. For purely elastic collisions (where the total kinetic energy is conserved), the maximum transferred energy $E_{\max}$ corresponds to electron backscattering, that is, $\theta = \pi$. However, when the impacted atom is moving, $E_{\max}$ will also depend on its speed.

To calculate the cross section, we use the approximation of McKinley and Feshbach[48] of the original series solution of Mott to the Dirac equation, which is very accurate for low-Z elements and sub-MeV beams. This gives the cross section as a function of the electron scattering angle as

$$\sigma(\theta) = \sigma_R \left[ 1 - \beta^2 \sin^2(\theta/2) + \pi \frac{Ze}{\hbar c} \beta \sin(\theta/2)(1 - \sin(\theta/2)) \right], \quad (15)$$

where $\beta = v/c$ is the ratio of electron speed to the speed of light (0.446225 for 60 keV electrons) and $\sigma_R$ is the classical Rutherford scattering cross section

$$\sigma_R = \left( \frac{Ze^2}{4\pi\epsilon_0 2m_0c^2} \right)^2 \frac{1 - \beta^2}{\beta^4} \csc^4(\theta/2). \quad (16)$$

Using equation 14 this can be rewritten as a function of the transferred energy[49] as

$$\sigma(E) = \left( \frac{Ze^2}{4\pi\epsilon_0 2m_0c^2} \frac{E_{\max}}{E} \right)^2 \frac{1 - \beta^2}{\beta^4} \left[ 1 - \beta^2 \frac{E}{E_{\max}} + \pi \frac{Ze^2}{\hbar c} \beta \left( \sqrt{\frac{E}{E_{\max}}} - \frac{E}{E_{\max}} \right) \right]. \quad (17)$$

**Distribution of atomic vibrations.** The maximum energy (in eV) that an electron with mass $m_e$ and energy $E_e = eU$ (corresponding to acceleration voltage $U$) can transfer to a nucleus of mass $M$ that is moving with velocity $v$ is

$$E_{\max}(v, E_e) = \frac{(r+t)^2}{2M} = \frac{\left( \sqrt{E_e(E_e + 2m_ec^2)} + Mvc + \sqrt{(E_e + E_n)(E + 2m_ec^2 + E_n)} \right)^2}{2Mc^2}, \quad (18)$$

where $r = \frac{1}{c}\sqrt{E_e(E_e + 2m_ec^2)} + Mv$ and $t = \frac{1}{c}\sqrt{(E_e + E_n)(E_e + 2m_ec^2 + E_n)}$ are the relativistic energies of the electron and the nucleus, and $E_n = Mv^2/2$ the initial kinetic energy of the nucleus in the direction of the electron beam.

The probability distribution of velocities of the target atoms in the direction parallel to the electron beam follows the normal distribution with a standard deviation equal to the temperature-dependent mean square velocity $\overline{v_z^2}(T)$,

$$P(v_z, T) = \frac{1}{\sqrt{2\pi \overline{v_z^2}(T)}} \exp\left( \frac{-v_z^2}{2\overline{v_z^2}(T)} \right). \quad (19)$$

**Total cross section with vibrations.** The cross section is calculated by numerically integrating equation 17 multiplied by the Gaussian velocity distribution (equation 19) over all velocities $v$ where the maximum transferred energy (equation 18) exceeds the displacement threshold energy $T_d$:

$$\sigma(T, E_e) = \int_{E_{\max}(v, E_e) \geq T_d} P(v, T) \sigma(E_{\max}(v, E_e)) dv \quad (20)$$

$$= \int_0^{v_{\max}} \frac{1}{\sqrt{2\pi \overline{v_z^2}(T)}} \exp\left( \frac{-v^2}{2\overline{v_z^2}(T)} \right) \left( \frac{Ze^2}{4\pi\epsilon_0 2m_0c^2} \frac{E_{\max}(v, E_e)}{E} \right)^2 \frac{1 - \beta^2}{\beta^4}$$

$$\left[ 1 - \beta^2 \frac{E}{E_{\max}(v, E_e)} + \pi \frac{Ze^2}{\hbar c} \beta \left( \sqrt{\frac{E}{E_{\max}(v, E_e)}} - \frac{E}{E_{\max}(v, E_e)} \right) \right] \quad (21)$$

$$\Theta[E_{\max}(v, E_e) - E_d] dv,$$

where $E_{\max}(v, E_e)$ is given by equation 18, the term $\Theta[E_{\max}(v, E_e) - E_d]$ is the Heaviside step function, $T$ is the temperature and $E_e$ is the electron kinetic energy.

The upper limit for the numerical integration $v_{\max} = 8\sqrt{\overline{v_z^2}}$ was chosen so that the velocity distribution is fully sampled.

**Displacement threshold simulation.** For estimating the displacement threshold energy, we used DFT molecular dynamics as established in our previous studies[12,13,50,51]. The threshold was obtained by increasing the initial kinetic energy

of a target atom until it escaped the structure during the molecular dynamics run. The calculations were performed using the grid-based projector-augmented wave code (GPAW), with the computational grid spacing set to 0.18 Å. The molecular dynamics calculations employed a double zeta linear combination of atomic orbitals basis[52] for a $8 \times 6$ unit cell of 96 atoms, with a $5 \times 5 \times 1$ Monkhorst-Pack **k**-point grid[53] used to sample the Brillouin zone. A timestep of 0.1 fs was used for the Velocity-Verlet dynamics[54], and the velocities of the atoms initialized by a Maxwell–Boltzmann distribution at 50 K, equilibrated for 20 timesteps before the simulated impact.

To describe exchange and correlation, we used the local density approximation[41], and the Perdew-Burke-Ernzerhof (PBE)[55], Perdew-Wang 1991 (PW91, ref. 41), RPBE[56] and revPBE[57] functionals, yielding displacement threshold energies of 23.13, 21.88, 21.87, 21.63 and 21.44 eV (these values are the means of the highest simulated kinetic energies that did not lead to an ejection and the lowest that did, respectively). Additionally, we tested the C09 (ref. 21) functional to see whether inclusion of the van der Waals interaction would affect the results. This does bring the calculated threshold energy down to [21.25, 21.375] eV, in better agreement with the experimental fit. However, a more precise algorithm for the numerical integration of the equations of motion, more advanced theoretical models for the interaction, or time-dependent DFT may be required to improve the accuracy of the simulations further.

**Varying mean square velocity with concentration.** Since the phonon dispersion of isotopically mixed graphene gives a slightly different out-of-plane mean square velocity for the atomic vibrations, for calculating the cross section for each concentration, we assumed the velocity of mixed concentration areas to be linearly proportional to the concentration

$$v_{\mathrm{mix}} = c v_{12} + (1-c) v_{13}, \tag{22}$$

where $c$ is the concentration of $^{12}C$ and $v_{12/13}$ are the atomic velocities for normal and heavy graphene, respectively.

**Raman spectroscopy.** A Raman spectrometer (NT MDT Ntegra Spectra) equipped with a 532 nm excitation laser was used for Raman measurements. A computer-controlled stage allowed recording a Raman spectrum map over the precise hole on which the electron microscopy measurements were conducted, which was clearly identifiable from neighboring spot contamination and broken film holes.

The frequencies $\omega$ of the optical phonon modes vary with the atomic mass $M$ as $\omega \propto M^{-1/2}$ due to the mass prefactor of the dynamical matrix. This makes the Raman shifts of $^{13}C$ graphene $(12/13)^{-1/2}$ times smaller, allowing the mapping and localization of $^{12}C$ and $^{13}C$ domains[28] with a spatial resolution limited by the size of the laser spot (nominally $\sim 400$ nm). The shifts of the G and 2D bands compared with a corresponding normal graphene sample are given by $\omega(c) = \omega_{12}\left[1 - \sqrt{\frac{12 + c_0^{13}}{12 + (1-c)}}\right]$, where $\omega_{12}$ is the G (2D) line frequency of the normal sample, $c_0^{13} = 0.01109$ is the natural abundance of $^{13}C$, and $c$ is the unknown concentration of $^{12}C$ in the measured spot.

Due to background signal arising from the carbon support film of the TEM grid, we analyzed the shift of the 2D band, where two peaks were in most locations present in the spectrum. However, in many spectra these did not correspond to either fully $^{12}C$ or $^{13}C$ graphene[58], indicating isotope mixing within the Raman coherence length. To assign a single value to the $^{12}C$ concentration for the overlay of Fig. 3, we took into account both the shifts of the peaks (to estimate the nominal concentration for each signal) and their areas (to estimate their relative abundances) as follows:

$$\begin{aligned} c_{12}^{\mathrm{total}} &= c_{12}^{\mathrm{A}} \frac{A}{A+B} + c_{12}^{\mathrm{B}} \frac{B}{A+B} \\ &= \left(1 - \frac{\omega_{\mathrm{A}} - \omega_{12}}{\omega_{12} - \omega_{13}}\right) \frac{A}{A+B} + \frac{\omega_{\mathrm{B}} - \omega_{13}}{\omega_{12} - \omega_{13}} \frac{B}{A+B}, \end{aligned} \tag{23}$$

where $c_{12}^{\mathrm{A/B}}$ are the nominal concentrations of $^{12}C$ determined from the measured higher and lower 2D Raman shift peak positions, $\omega_{\mathrm{A/B}}$ are the measured peak centers of the higher and lower 2D signals, and A and B are their integrated intensities. The peak positions of fully $^{12}C$ and $^{13}C$ graphene were taken from the highest and lowest peak positions in the entire mapped area (covering several dozen Quantifoil holes), giving $\omega_{12} = 2,690 \, \mathrm{cm}^{-1}$ and $\omega_{13} = 2,600 \, \mathrm{cm}^{-1}$. The fitted 2D spectra, arranged in the same $6 \times 6$ grid as the overlay, can be found as Supplementary Fig. 3

**Data availability.** The full STEM time series data on which the determination of the $^{12}C$ and $^{13}C$ displacement cross sections (Fig. 2) are based are available on *figshare* with the identifier http://dx.doi.org/10.6084/m9.figshare.c.3311946 (ref. 20). The STEM data of Fig. 3 are available upon request. All other data are contained within the article and its Supplementary Information files.

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

## Acknowledgements

T.S. acknowledges funding from the Austrian Science Fund (FWF) via project P 28322-N36, and the computational resources of the Vienna Scientific Cluster. J.K. acknowledges funding from the Wiener Wissenschafts-, Forschungs- und Technologiefonds (WWTF) via project MA14-009. C.H., G.A., C.M. and J.C.M. acknowledge funding by the European Research Council Grant No. 336453-PICOMAT. T.J.P. was supported by the European Union's Horizon 2020 research and innovation programme under the Marie Skłodowska-Curie grant agreement No. 655760-DIGIPHASE. We further thank Ondrej Krivanek for useful feedback.

## Author contributions

T.S. performed theoretical and statistical analyses and DFT simulations, participated in STEM experiments and their analysis, and drafted the manuscript. C.H. performed sample synthesis, and participated in STEM experiments, their analysis, and the Raman analysis. G.A. participated in sample synthesis and the Raman analysis. G.T.L. participated in STEM experiments and their analysis. C.M. and T.J.P. prepared special alignments for the STEM instrument with J.K., who supervised the theoretical and statistical analyses and STEM experiments. J.K. and J.C.M conceived and supervised the study.

## Additional information

**Competing financial interests:** T.S., J.C.M. and J.K. are named on a patent application relating to this method of isotope analysis (application number EP16183371). All other authors declare no competing financial interests.

DOI: 10.1038/ncomms15780    **OPEN**

# Corrigendum: Isotope analysis in the transmission electron microscope

Toma Susi, Christoph Hofer, Giacomo Argentero, Gregor T. Leuthner, Timothy J. Pennycook, Clemens Mangler, Jannik C. Meyer & Jani Kotakoski

*Nature Communications* 7:13040 doi: 10.1038/ncomms13040 (2016); Published 10 Oct 2016; Updated 30 Aug 2017

This Article contains typographical errors in Fig. 3 and Equation 23.

In Fig. 3c, the labels '$^{12}$C' and '$^{13}$C' in the key should be reversed. The correct version of Fig. 3 appears below as Fig. 1.

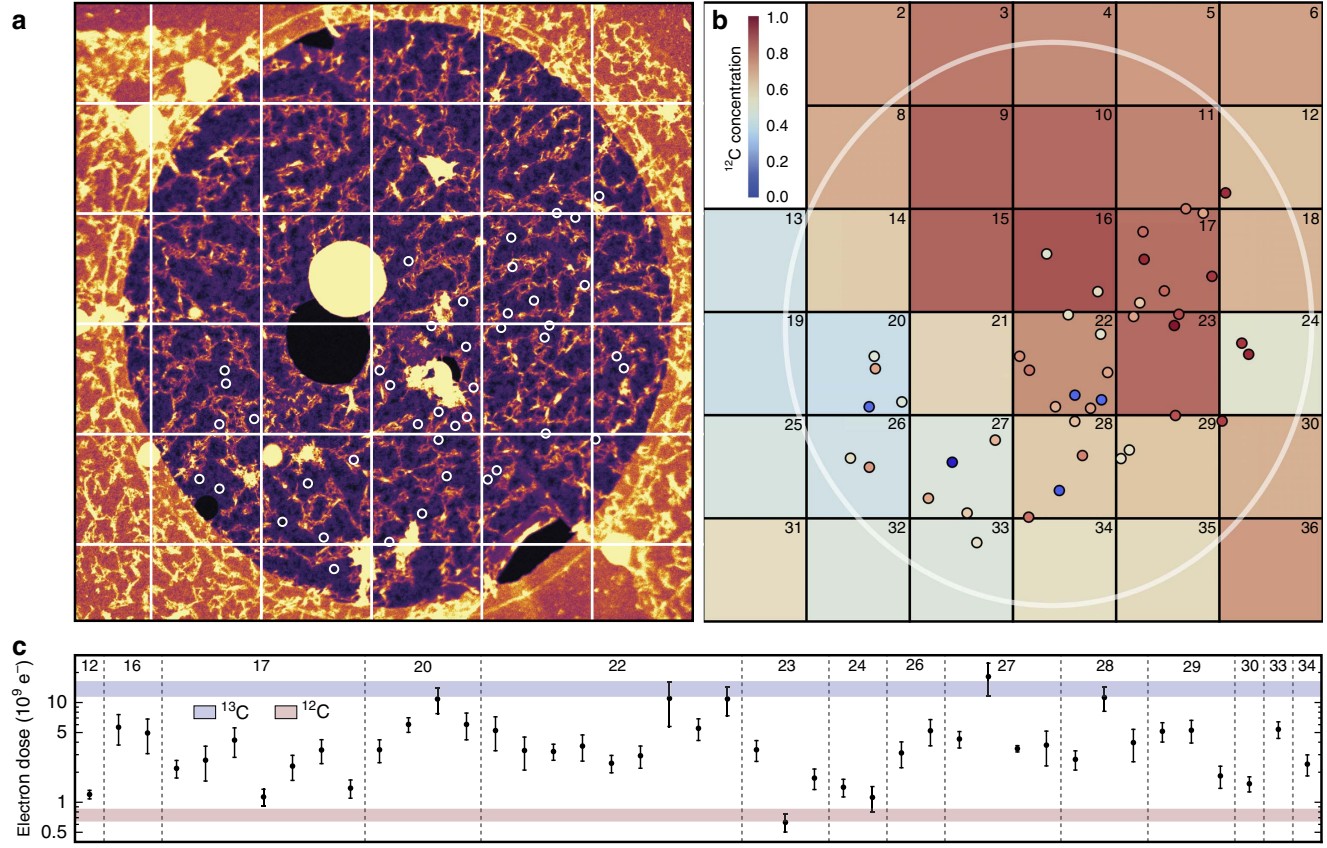

**Figure 1**

The correct form of Equation 23 is as follows:

$$c_{12}^{total} = c_{12}^{A}\frac{A}{A+B} + c_{12}^{B}\frac{B}{A+B} = \left(1 - \frac{\omega_{12}-\omega_A}{\omega_{12}-\omega_{13}}\right)\frac{A}{A+B} + \frac{\omega_B-\omega_{13}}{\omega_{12}-\omega_{13}}\frac{B}{A+B}$$

