## [Peer review file · Nature Communications]

Reviewers' comments:

Reviewer #1 (Remarks to the Author):

Susi et al. have shown that by measuring the atomic displacement of individual C atoms in monolayer graphene via a combination of scanning transmission electron microscopy (STEM) imaging, statistical physics and density functional theory (DFT) is possible to distinguish their isotope identity (i.e., ^{12}C and ^{13}C). Their STEM measurements are corroborated with Raman spectroscopy experiments.

I find their approach to the problem quite clever and original. The text is generally well written and easy to follow. The figures are good, although some improvements are required (see below).

In general I enjoyed reading the manuscript and I think the approach presented here to study isotopes with STEM is going to be of interest not only to the microscopy community but also to materials scientist interested in 2D materials.

I think their conclusions are fair, they cite previous work fairly (a criticism on this point below).

I think the manuscript will be of interest for a broad audience and could be published in Nat. Comm after some corrections.

Major criticisms:

1. Title. I know that the authors try to have a general title that is catchy. Yet, I think the title is a bit too general. Wouldn't be better to have a title that directly tells a possible reader what the authors did, such as "Isotope analysis on the atomic scale via scanning transmission electron microscopy"?

2. Figures.

2.a. Figure 3 is not very clear to understand. I think that the main reason is that the results of Figure 3 are linked with the results of Sup Figures 3 and 4. However, when reading the manuscript I had to jump to the supplementary information before I could go to Figure 3.

I suggest the authors to move Sup figures 3 & 4 to the main text. The figures need to be improved and captions modified if that is the case. For instance, for the case of Sup Figure 3, the authors can chose only 3 presentative panels for the Raman measurements and their respective ^{12}C concentration (and have the rest of the data in the supplementary section). In that way the reader can follow the methodology better within the main text of the manuscript.

3. Citation

3.a. In the section "Interaction cross section", page 15, the authors say that "As noted by Meyer and co workers,11..." and then write down Eq 12 that correlates the energy transferred to an atomic nuclei by a fast electron. That Eq is not originally derived by Meyer and coworkers!!! It is a well known equation that can be found in classical books, for instance see "Theory of Atomic Collisions by N. F. Mott H. S. W. Massey (1965)".

So I suggest that the authors to amend the text because it gives the impression to a possible reader that Meyer and co-workers came out with the relationship presented in eq 12, which is far from true!

Minor suggestions

1. In the figure caption of Sup. Fig 3. the text refers to "Figure 4" of the main manuscript. There are only a total of three figures in the main text. I think the authors meant Figure 3, is that right?

Reviewer #2 (Remarks to the Author):

This is an interesting paper that is clearly written and in which the experimental data is clearly presented.

The essence of the paper is the separation of isotopes in the microscope based on differential energy thresholds for sputtering. This is a novel idea which offers new possibilities that extend the range of structural data in the microscope. To the best of my knowledge this is the first time the STEM has been used to probe mass rather than atomic number which, in my view, gives the paper the novelty required for publication in a NPG Journal

Nonetheless there are a number of technical issues that I would like the authors to comment on:

1. P2. The authors describe energy transfer from elastic backscattering. I understand their reasoning but feel that the terminology is not correct - an elastic process is strictly defined as one in which no energy is transferred.
2. P3 How does the calculated band structure calculated by DFT compare to the experimentally measured band structure. This would seem to be important in any treatment of errors as the isotopic differences are relatively small.
3. Fig 1. and corresponding text. I am surprised that in the experiments described the mono vacancy does not reconstruct as has been previously reported. Perhaps the authors could comment on this ?

The authors also hint at the issue of vacancy refilling and it would be useful to have their views on how this might affect the results reported, particularly since figure suggests that there is significant contamination / amorphous carbon on the as prepared sample.

4. Methods. The supercell used for the phonon dispersion calculations (7 x 7) seems small.

Minor Corrections:

1. Some of the latter figures S2 and S4 in particular could be clearer.

Reviewer #3 (Remarks to the Author):

See attached pdf file (below).

Referee report on: **Isotope Analysis on the Atomic Scale**

T. Susi, C. Hofer, G. Argentero, G. T. Leuthner, T. J. Pennycook, C. Mangler, J. C. Meyer, and J. Kotakoski

The authors report a new method of determining isotope ratios at surfaces using a measurement of the electron dose required to emit an atom from the surface. They develop the underlying theory of the method, together with a sound statistical analysis of the data that enables the isotope ratio to be determined.

This is a very nice and clearly presented piece of work which demonstrates clearly that mass information can be extracted from electron microscopy data. I believe the work will be of interest to a wide audience, and is suitable for publication in Nature Communications with some minor corrections and clarifications to the manuscript. These are listed below.

1. At present, the abstract does not describe the contents of the manuscript until the very end. The authors should edit the abstract to state clearly at the beginning what they have achieved (e.g. beginning at the sentence “We show how the quantum mechanical description of lattice vibrations...”), ideally with a quantitative statement about the spatial resolution and accuracy/precision of their isotope measurements. The background information about other methods (i.e. from the start of the abstract up to “...mass spectrometer in a microscope”) should be moved to the start of the main paper as an introduction.
2. Since Nature Communications is intended for a fairly general scientific audience, rather than a specialist electron microscopy readership, there are one or two places in which definitions should be given. For example, the first time ‘dose’ is used on page 4, ‘electron dose’ would be more helpful for the uninitiated reader. Similarly, the ZA and ZO phonon modes should be defined the first time they appear in the text on page 3 (these are defined in the supplementary material, but not in the main article).
3. The fit between the DFT calculations and the experimental data is qualitatively good, reproducing the general energy dependence of the cross section. This indicates that the model captures the basic physics of the ejection process. However, the calculations do not reproduce the experimental data quantitatively within experimental error, so saying that “the theory accurately describes the contribution of vibrations” (page 5) is perhaps something of an overstatement. The authors should clarify this within the text, and discuss possible reasons for the discrepancy, if these are known.
4. Figure 3 summarises the results of the isotope analysis. However, the results shown in the figure can barely be seen. The figure needs to be made much clearer, either by altering the contrast between the image and the superimposed data, increasing the size of the superimposed data points (even if this is at the expense of an accurate scale for the superimposed data), or a combination of the two.
5. The paper lacks a suitably detailed comparison of the new method with the state of the art in the field. This does not need to be long, but I think that including such a discussion would greatly strengthen the manuscript and put the work into context. The discussion should also include a clear statement of the spatial resolution and the accuracy and precision of the isotope concentration measurement, ideally comparing these with the same parameters for other methods.

There are also one or two minor typos and places where minor rewordings would improve clarity. These are listed below:

1. In the second sentence of the abstract, "...graphene, where..." would be better as "...graphene, in which..."
2. On page 4, "...recording at room temperature time series using..." would be better as "...recording time series at room temperature using..."
3. At the start of the last paragraph on page 4, "...experimental data set where a clear..." would be better as "...experimental data set within which a clear..."
4. At the bottom of page 5, "This is understandable by the vastly shorter lifetimes..." would be better as "This can be understood in terms of the very short lifetimes..."
5. At the top of page 7, "...concentration responsible for such a dose" would be better as "...concentration corresponding to the measured threshold dose". At the end of this paragraph, "...far from atomically sharp, but can span hundreds of nm where the carbon..." would be better as "...far from atomically sharp, instead spanning a region of hundreds of nanometres, within which the carbon..."
6. In the caption to figure 1, second to last line, "aradius" should be "a radius". It might also be worth writing out "pixels" in full, as not everyone will be familiar with the abbreviation "px".
7. In the caption to Figure 2, "a error weighted" should be "an error-weighted"

Reviewer #1 (Remarks to the Author):

Susi et al. have shown that by measuring the atomic displacement of individual C atoms in monolayer graphene via a combination of scanning transmission electron microscopy (STEM) imaging, statistical physics and density functional theory (DFT) is possible to distinguish their isotope identity (i.e., ^{12}C and ^{13}C). Their STEM measurements are corroborated with Raman spectroscopy experiments.

I find their approach to the problem quite clever and original. The text is generally well written and easy to follow. The figures are good, although some improvements are required (see below).

In general I enjoyed reading the manuscript and I think the approach presented here to study isotopes with STEM is going to be of interest not only to the microscopy community but also to materials scientist interested in 2D materials.

I think their conclusions are fair, they cite previous work fairly (a criticism on this point below).

I think the manuscript will be of interest for a broad audience and could be published in Nat. Comm after some corrections.

We are pleased that the referee enjoyed the manuscript and finds our approach to be original. The points of criticism are likewise well taken, and we have addressed them as described below.

Major criticisms:

1. Title. I know that the authors try to have a general title that is catchy. Yet, I think the title is a bit too general. Wouldn't be better to have a title that directly tells a possible reader what the authors did, such as "Isotope analysis on the atomic scale via scanning transmission electron microscopy"?

This is a valid point; although we spent considerable effort in choosing the simplest yet still descriptive title, perhaps it did become too general. After some further thought, we believe the title "Isotope analysis in the transmission electron microscope" is a good compromise. Atomic scale is implied in transmission electron microscopy, and although we used a scanning instrument, there is no fundamental reason why a parallel illumination TEM would not work.

2. Figures.

2.a. Figure 3 is not very clear to understand. I think that the main reason is that the results of Figure 3 are linked with the results of Sup Figures 3 and 4. However, when reading the manuscript I had to jump to the supplementary information before I could go to Figure 3. I suggest the authors to move Sup figures 3 & 4 to the main text. The figures need to be improved and captions modified if that is the case. For instance, for the case of Sup Figure 3, the authors can chose only 3 presentative panels for the Raman measurements and their respective ^{12}C concentration (and have the rest of the data in the supplementary section). In that way the reader can follow the methodology better within the main text of the manuscript.

Figure 3 was admittedly not optimally designed, and the suggestion to combine data from the supplementary figures is a good idea. We have thus completely reworked Figure 3, which now should be easier to understand. However, we do think that adding Raman spectra to it would make it unnecessarily complicated: the determination of isotope concentration from the Raman 2D band is well established and thus the data itself is shown only for completeness.

3. Citation

3.a. In the section "Interaction cross section", page 15, the authors say that "As noted by Meyer and co workers, 11..." and then write down Eq 12 that correlates the energy transferred to an atomic nuclei by a fast electron. That Eq is not originally derived by Meyer and coworkers!!! It is a well known equation that can be found in classical books, for instance see "Theory of Atomic Collisions by N. F. Mott H. S. W. Massey (1965)".

So I suggest that the authors to amend the text because it gives the impression to a possible reader that Meyer and co-workers came out with the relationship presented in eq 12, which is far from true!

We did not mean to imply Meyer and co-workers derived this, but we can see how the citation could give that impression. We have clarified this and further cited the suggested textbook.

Minor suggestions

1. In the figure caption of Sup. Fig 3. the text refers to "Figure 4" of the main manuscript. There are only a total of three figures in the main text. I think the authors meant Figure 3, is that right?

Indeed; now fixed.

Reviewer #2 (Remarks to the Author):

This is an interesting paper that is clearly written and in which the experimental data is clearly presented.

The essence of the paper is the separation of isotopes in the microscope based on differential energy thresholds for sputtering. This is a novel idea which offers new possibilities that extend the range of structural data in the microscope. To the best of my knowledge this is the first time the STEM has been used to probe mass rather than atomic number which, in my view, gives the paper the novelty required for publication in a NPG Journal.

We thank the referee for the positive evaluation, and address their astute technical comments below. We would like to clarify one minor point, though: the *threshold energies* for sputtering are identical for isotopes of the same element, what are differential are the *cross sections*, which depend on the mass.

Nonetheless there are a number of technical issues that I would like the authors to comment on:
1. P2. The authors describe energy transfer from elastic backscattering. I understand their reasoning but feel that the terminology is not correct - an elastic process is strictly defined as one in which no energy is transferred.

In conventional usage, there is a distinction between elastic *collisions* and elastic *scattering*: in the former, the total kinetic energy of the system needs to be conserved, whereas in the latter—which seems to be what the referee is thinking of—the kinetic energies of each participating particle need to be conserved. We were thinking of the first definition, which does apply to pure electrostatic interactions, including in our case the ~20 eV energy transferred from the impinging electron to the displacing nucleus.

To avoid the ambiguity the referee points out, we have reformulated the sentences including the first instance of “elastic” in the manuscript to read:

When a highly energetic electron is scattered by the electrostatic potential of an atomic nucleus, a maximal amount of kinetic energy (inversely proportional to the mass of the nucleus, $\propto 1/M$) can be transferred when the electron backscatters.

In the second instance, we have explicitly defined what we mean:

For purely elastic collisions (where the total kinetic energy is conserved), the maximum transferred energy E_{\max} corresponds to electron backscattering, i.e. $\theta = \pi$.

2. P3 How does the calculated band structure calculated by DFT compare to the experimentally measured band structure. This would seem to be important in any treatment of errors as the isotopic differences are relatively small.

As we mentioned in the manuscript, the calculated phonon band structure is in excellent agreement with both earlier simulations and experiments, see e.g. Figure 3 of Wirtz et al. 2004 (doi:10.1016/j.ssc.2004.04.042) where they compare their practically identical phonon dispersion to several experimental techniques, or the high-quality inelastic x-ray scattering data of Mohr et al. 2007 (doi:10.1103/PhysRevB.76.035439). In our view, the phonon band structure is already so well established that it did not warrant a closer comparison in our work, where the decomposition into in-plane and out-of-plane components was its main use. Furthermore, since the phonon band structure is identical for both ^{12}C and ^{13}C graphene apart from a mass ratio scaling factor, any minor inaccuracies in the calculation would not affect our results.

3. Fig 1. and corresponding text. I am surprised that in the experiments described the mono vacancy does not reconstruct as has been previously reported. Perhaps the authors could comment on this ?

The relevant figure is panel f of Figure 1 showing the atomic vacancy created by a displacement. Although the image contrast does indeed appear to show an unreconstructed vacancy, this is very likely due to the influence of the electron irradiation. In TEM, the image contrast is formed from an average of atomic positions over the frame acquisition time (typically 1 s or less). If one of the three equivalent Jahn-Teller reconstructions is favored for example due to strain, the image of vacancy will show a reconstructed C–C bond. However, in STEM, the frame is not simultaneously exposed, but rather scanned line by line. In this case, the approaching electron beam can break the rather weak reconstructed bond before it contributes to the image contrast, resulting in the image of an unreconstructed vacancy.

The authors also hint at the issue of vacancy refilling and it would be useful to have their views on how this might affect the results reported, particularly since figure suggests that there is significant contamination / amorphous carbon on the as prepared sample.

Significant carbon residue indeed appears on the graphene lattice, as is the case in almost all samples. This provides a source of mobile C adatoms that can fill created atomic vacancies. In earlier work on this topic, the method that was used to estimate displacement thresholds from TEM measurements was the counting of vacancies in a large field of view as a function of time. This measurement is especially sensitive to vacancy filling, being very likely responsible for the underestimate of the old 100 kV ^{12}C datapoint, as mentioned in the manuscript.

Our experiments were specifically designed to minimize the effect of vacancy refilling. For this purpose, we used very small fields of view, allowing us to scan each frame as fast as possible while still distinguishing each atom in every frame. Thus although we cannot be absolutely sure that our dataset contains no undetected immediately refilled displacements, based on our Poisson analysis we are confident they do not significantly affect our results.

4. *Methods. The supercell used for the phonon dispersion calculations (7 x 7) seems small.*

Although 7 x 7 might seem small, in the frozen phonon calculation the important consideration is whether the forces on distant atoms due to the displacement of the two unit cell atoms are negligible or not. We double-checked this and confirmed that the forces at the edge of the cell are 100-1000 times smaller than on the displaced atoms, which indicates that their contribution to the dynamical matrix will not be important. Indeed, previous calculations found in the literature have used 6 x 6 (10.1103/PhysRevB.77.125401) or even smaller cells (10.1103/PhysRevB.76.035439).

Minor Corrections:

1. *Some of the latter figures S2 and S4 in particular could be clearer.*

Thank you for pointing this out. We have strived to improve the clarity of all the manuscript figures.

Reviewer #3 (Remarks to the Author):

The authors report a new method of determining isotope ratios at surfaces using a measurement of the electron dose required to emit an atom from the surface. They develop the underlying theory of the method, together with a sound statistical analysis of the data that enables the isotope ratio to be determined.

This is a very nice and clearly presented piece of work which demonstrates clearly that mass information can be extracted from electron microscopy data. I believe the work will be of interest to a wide audience, and is suitable for publication in Nature Communications with some minor corrections and clarifications to the manuscript. These are listed below.

We are grateful for this kind evaluation and especially for the detailed feedback that we address below.

At present, the abstract does not describe the contents of the manuscript until the very end. The authors should edit the abstract to state clearly at the beginning what they have achieved (e.g. beginning at the sentence “We show how the quantum mechanical description of lattice vibrations...”), ideally with a quantitative statement about the spatial resolution and accuracy/precision of their isotope measurements. The background information about other methods (i.e. from the start of the abstract up to “...mass spectrometer in a microscope”) should be moved to the start of the main paper as an introduction.

The abstract was admittedly written for a letter format, where it was required to cover the introduction as well. However, the referee is right to point out that the abstract is no longer appropriate for a full article. We have thus moved the earlier abstract to be the first paragraph of the introduction, and written an entirely new abstract that addresses these points better.

Since Nature Communications is intended for a fairly general scientific audience, rather than a specialist electron microscopy readership, there are one or two places in which definitions should be given. For example, the first time ‘dose’ is used on page 4, ‘electron dose’ would be more helpful for the uninitiated reader. Similarly, the ZA and ZO phonon modes should be defined the first time they appear in the text on page 3 (these are defined in the supplementary material, but not in the main article).

Very good point; these have now been explicitly defined.

The fit between the DFT calculations and the experimental data is qualitatively good, reproducing the general energy dependence of the cross section. This indicates that the model captures the basic physics of the ejection process. However, the calculations do not reproduce the experimental data quantitatively within experimental error, so saying that “the theory accurately describes the contribution of vibrations” (page 5) is perhaps something of an overstatement. The authors should clarify this within the text, and discuss possible reasons for the discrepancy, if these are known.

After submitting the manuscript, the accuracy of the calculations continued to vex us. We thus performed additional calculations using a van der Waals exchange-correlation functional, and were able to bring the best computational estimate down to [21.25, 21.375] eV with the C09 functional. The corresponding curve now passes through practically all the experimental error bars (Figure 2 has been updated), although clearly there is further room for improvement.

We believe the reason for the discrepancy are numerical errors in the integration of the equations of motion and the inherent sensitivity of the trajectory to minor variations in the forces on the displaced atom. During the ejection, very small inaccuracies in the force at each location of the trajectory compound, resulting in a noticeable difference in the displacement threshold energy. The quality of the experimental data has now reached a level where more accurate theoretical tools may be needed, and we have added an explicit mention of this to the manuscript.

Figure 3 summarises the results of the isotope analysis. However, the results shown in the figure can barely be seen. The figure needs to be made much clearer, either by altering the contrast between the image and the superimposed data, increasing the size of the superimposed data points (even if this is at the expense of an accurate scale for the superimposed data), or a combination of the two.

Figure 3 clearly needed another kind of design to be clear. We have thus completely reworked it, including both a clearer overlay on the STEM image and another panel with the color-coded concentration information next to it. We have also included the dose data from the Supplement, now assigned to the same grid squares as the top panels. We believe this is a significantly better way to represent the data and we thank the referee for their criticism of the original figure.

The paper lacks a suitably detailed comparison of the new method with the state of the art in the field. This does not need to be long, but I think that including such a discussion would greatly strengthen the manuscript and put the work into context. The discussion should also include a clear statement of the spatial resolution and the accuracy and precision of the isotope concentration measurement, ideally comparing these with the same parameters for other methods.

This is a good suggestion. Although it is not entirely straightforward to compare our technique to established mass analysis methods, briefly surveying the state of the art and contrasting with it certainly can improve the paper. We have added a section to address this before the conclusions.

There are also one or two minor typos and places where minor rewordings would improve clarity. These are listed below:

- 1. In the second sentence of the abstract, "...graphene, where..." would be better as "...graphene, in which..."*
- 2. On page 4, "...recording at room temperature time series using..." would be better as "...recording time series at room temperature using..."*
- 3. At the start of the last paragraph on page 4, "...experimental data set where a clear..." would be better as "...experimental data set within which a clear..."*
- 4. At the bottom of page 5, "This is understandable by the vastly shorter lifetimes..." would be better as "This can be understood in terms of the very short lifetimes..."*
- 5. At the top of page 7, "...concentration responsible for such a dose" would be better as "...concentration corresponding to the measured threshold dose". At the end of this paragraph, "...far from atomically sharp, but can span hundreds of nm where the carbon..." would be better as "...far from atomically sharp, instead spanning a region of hundreds of nanometres, within which the carbon..."*
- 6. In the caption to figure 1, second to last line, "aradius" should be "a radius". It might also be worth writing out "pixels" in full, as not everyone will be familiar with the abbreviation "px".*
- 7. In the caption to Figure 2, "a error weighted" should be "an error-weighted"*

Many thanks for this detailed reading; all of these have now been corrected.

REVIEWERS' COMMENTS:

Reviewer #2 (Remarks to the Author):

The authors response to my earlier review comprehensively addresses all my technical questions raised.

Hence I think that this is an important paper which should be published.

My only comment is that the author responses to my original points 2, 3 and 4 add value to the paper and should be incorporated into either the main text or the supporting or extended information.

Reviewer #3 (Remarks to the Author):

The authors have responded more than satisfactorily to all of my comments on the original manuscript, and I am very happy to recommend publication of the revised manuscript in Nature Communications.

REVIEWERS' COMMENTS:

Reviewer #2 (Remarks to the Author):

The authors response to my earlier review comprehensively addresses all my technical questions raised.

Hence I think that this is an important paper which should be published.

My only comment is that the author responses to my original points 2, 3 and 4 add value to the paper and should be incorporated into either the main text or the supporting or extended information.

We are pleased that our replies satisfied all of the points the referee raised. Following their suggestion, we have incorporated the substance of our answers to their comments in the needed places.

Reviewer #3 (Remarks to the Author):

The authors have responded more than satisfactorily to all of my comments on the original manuscript, and I am very happy to recommend publication of the revised manuscript in Nature Communications.

We thank the referee for their time and their valuable feedback.